# A simple model considering spiking probability during extracellular axon stimulation

**Frank Rattay**⊙*[⊙], **Thomas Tanzer**[⊙]

Institute of Analysis and Scientific Computing, Vienna University of Technology, Vienna, Austria

⊙ These authors contributed equally to this work.
* frank.rattay@tuwien.ac.at

**Data Availability Statement:** All relevant data are within the manuscript.

**Funding:** This work was supported by the Austrian Science Fund (FWF), grant no. 29650. The fund supported publication costs but the authors received no specific funding for this work. The

## Abstract

The spiking probability of an electrically stimulated axon as a function of stimulus amplitude increases in a sigmoidal dependency from 0 to 1. However, most computer simulation studies for neuroprosthetic applications calculate thresholds for neural targets with a deterministic model and by reducing the sigmoid curve to a step function, they miss an important information about the control signal, namely how the spiking efficiency increases with stimulus intensity. Here, this spiking efficiency is taken into account in a compartment model of the Hodgkin Huxley type where a noise current is added in every compartment with an active membrane. A key parameter of the model is a common factor knoise which defines the ion current fluctuations across the cell membrane for every compartment by its maximum sodium ion conductance. In the standard model Gaussian signals are changed every 2.5 μs as a compromise of accuracy and computational costs. Additionally, a formula for other noise transmission times is presented and numerically tested. Spiking probability as a function of stimulus intensity can be approximated by the cumulative distribution function of the normal distribution with RS = $\sigma/\mu$. Relative spread RS, introduced by Verveen, is a measure for the spread (normalized by the threshold intensity $\mu$), that decreases inversely with axon diameter. Dynamic range, a related measure used in neuroprosthetic studies, defines the intensity range between 10% and 90% spiking probability. We show that (i) the dynamic range normalized by threshold is 2.56 times RS, (ii) RS increases with electrode—axon distance and (iii) we present knoise values for myelinated and unmyelinated axon models in agreement with recoded RS data. The presented method is applicable for other membrane models and can be extended to whole neurons that are described by multi-compartment models.

## Introduction

Axons originate from the cell bodies of nerve cells as long extensions. They transmit binary neural signals (all or nothing) as action potentials (APs), which are short changes of the transmembrane voltage in the order of 1 ms and 100 mV. Experiments and theoretical

funders had no role in study design, data collection and analysis, decision to publish, or preparation of the manuscript.

**Competing interests:** The authors have declared that no competing interests exist.

investigations demonstrate that the axon is the most excitable part of a neuron for stimulation with microelectrodes as used in neuroprostheses [1, 2]. Therefore, most neural prostheses are designed to generate APs in the axons by electrical stimulation, e.g. for paralyzed, deaf or blind persons [3–6]. The challenge is to generate a neural pattern as a good supplement for a lost neural task.

Although, computer simulations are important tools to improve the generated patterns in the stimulated axons, most of them provide only data about the threshold current needed to generate an AP in a target neuron. However, for a selected axon a second information is important, too. There is an interval of the stimulus intensity where the probability to obtain an AP per stimulating pulse increases from 0 to 1. Unfortunately, quite few modeling studies include a stochastic component in order to replicate the noisiness of recorded spike patterns elicited by electrical stimulation, e. g. in cochlear implants [7–9]. Most modeling studies that include noise either refer to whole neurons, i.e., consisting of dendrites, soma, and axon [8, 9], or refer to a single excitable patch of membrane that does not interact with other sections of the neuron [7, 10, 11]. Here, we present a simple method to analyze the impact of parameters (e.g. axon type, axon diameter, electrode distance, duration and intensity of the stimulus pulse) on the probability that every stimulus pulse of an external microelectrode initiates an AP in an axon.

In comparison with a technical cable the conductivity of the intracellular fluid in the axon is very low and, second, the cell membrane is no perfect insulator. These two peculiarities cause a loss of the signal within a distance in the mm range if the signal is not amplified. Two types of axons exist, non-myelinated and myelinated ones. The non-myelinated axon has a uniform membrane with ion channels that enables the conductance of an AP with constant amplitude because the sodium and potassium ion currents across the active membrane compensate current leakage and excite the axon in front of the propagating AP. Myelinated axons have isolating sheaths of lipids around the axon. The myelin layer has short interruptions, called Nodes of Ranvier (NoR), where the conduced AP is amplified by the active cell membrane. As the myelinated 'internode' is much longer than the NoR myelinated fibers enjoy three major advantages over non-myelinated ones: (i) the conduction is faster, (ii) less energy is required, and (iii) fast conducting axons are thinner.

The understanding of how APs arise and are transmitted has been greatly influenced by the work of Hodgkin and Huxley and their famous model [12]. Sodium and potassium ion channels, which are opened or closed with a certain probability depending on the membrane voltage, are mainly responsible for the formation of an AP. Hodgkin and Huxley developed from experiments on non-myelinated axons a set of four differential equations that quantify the transmembrane currents during excitation (HH model). The state variables of the HH model are the transmembrane voltage $V_m$ and three gating variables (m, n, h) for the open probabilities of ion channels. However, these modeled open channel probabilities depend in a deterministic way on $V_m$ and thus models of the HH type miss the spiking irregularities seen in experiments (Fig 1).

Contrary to non-myelinated axons, in myelinated axons of warm blooded animals the potassium current is of negligible intensity [14]. An equation missed by the authors (Chiu, Ritchie, Rogart, and Stagg) was reconstructed by Sweeney in 1987 and consequently this model, which is of the HH type, was named CRRSS model [15–17]. The ion channel density in the nodes of Ranvier is an order larger than in non-myelinated axons. Consequently, for biomedical investigations, several modelers used the HH10 model with ten times enlarged HH conductances in the NoR, especially before the CRRSS model was available [18].

The knowledge based on models of the Hodgkin and Huxley type inspired researchers to build neural prostheses, where the electrically stimulated axon is the target to generate artificial

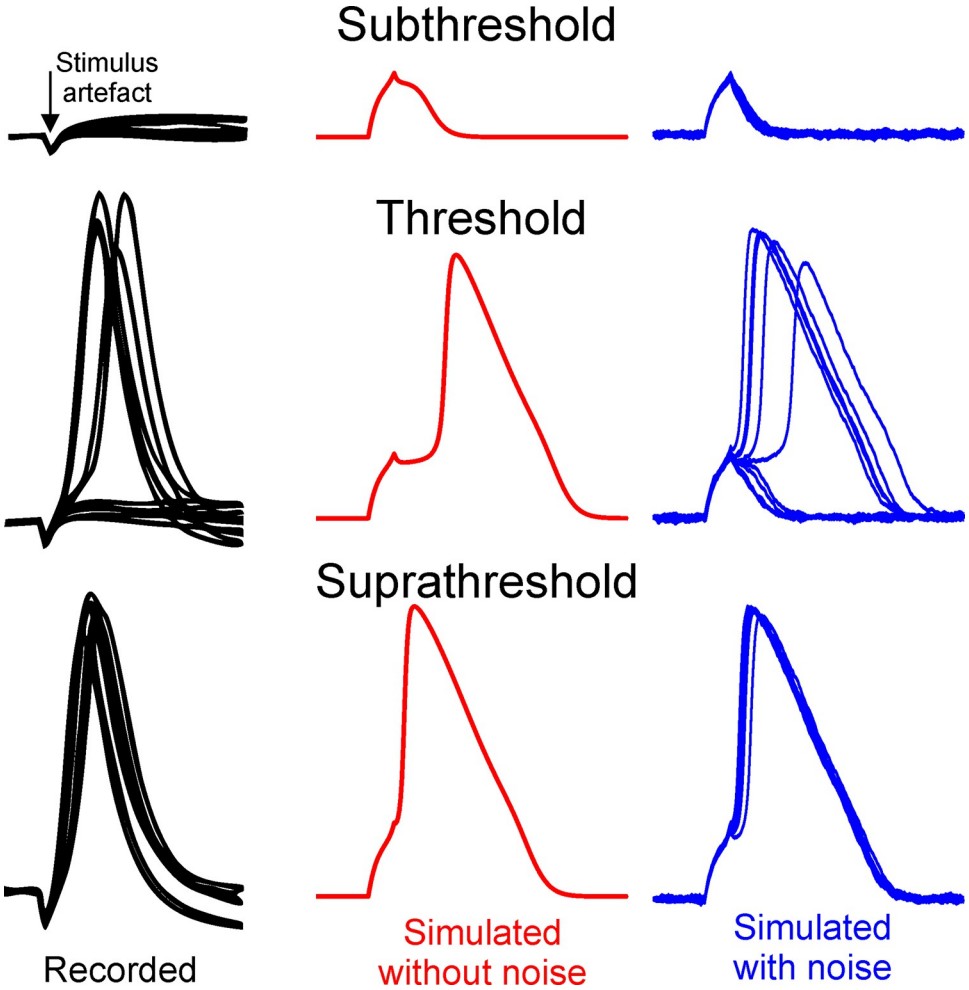

**Fig 1. Stochasticity in AP generation under constant stimulation conditions.** If 10 pulses are applied at threshold intensity 5 APs are expected but a repetition of the experiment may cause 4 or 7 APs. Stronger stimulation shows a precise response of one AP per pulse (bottom) but still some variation in the spike shapes. In an original model of the HH type (red examples) a noise term is added to include a stochastic behavior (blue). Recorded membrane voltages of retinal ganglion cells were redrawn from [13], simulations were calculated with the CRRSS model.

neural code. A successful example is the multi-electrode system of a cochlear implant. In a constant time-cycle each of the electrodes is active in a time slot and stimulates just axons of the auditory nerve from a frequency-specific region. This way the insertion depth of each of the electrodes is linked to a frequency section of the acoustic signal but the amplitude of the stimulating pulses controls the loudness of the related frequency band [7, 19]. In the natural situation loudness is coded by the spiking rate, APs per second, but in most of the implants the stimulating pulses have a constant cycle time. According to the classical HH model a specific axon will respond with an AP only if the stimulus intensity is above threshold (all or nothing) and thus it transfers in a target fiber either nothing for subthreshold stimulation or a constant loudness information defined by the cycle time. This is quite different to the natural code [20]. However, experiments in feline axons of the auditory nerve showed a sigmoidal increase of the spiking probability as a function of stimulus current [21] that defines a technical 'loudness control window' for a target fiber where the firing probability increases from 0.1 to 0.9. This stimulus window is called the dynamic range of the axon.

After evaluations of experiments Verveen recognized that firing probability as a function of stimulus intensity can be approximated by the cumulative distribution function of the normal distribution and titled the coefficient of variation of the approximating distribution the relative spread (RS) [22, 23]. RS is a dimensionless measure of stochasticity. The firing probability can be calculated by dividing the number of spikes by the number of stimulating pulses, where 100% firing probability means that every pulse generates a spike and 50% is called threshold. The 'threshold' (mean of the distribution function) as well as the 'spread' (standard deviation of the distribution function) depend on the stimulus duration, while the RS (quotient of spread and threshold) is independent of this stimulus duration for a relatively large interval of durations [10, 23, 24].

Neurons have intrinsic noisy dynamics due to random gating of ion channel populations, commonly referred to as "channel noise." This channel noise arises from the random opening and closing of ion channels embedded in the cell membrane. There are several ways to include a noise component in the deterministic Hodgkin Huxley model. The first variant is called current noise approach, where a stochastic term is added to the main equation of the HH model. Another possibility is to add the noise term to the equations describing the fraction of open subunits and the third common possibility for the HH model is called conductance noise where the noise term is introduced directly into the fractions of open channels [25–27].

Related to variant 1, Motz and Rattay [28] added a white noise signal of low amplitude in a local HH model to simulate electrical stimulation of the auditory nerve with sinusoidal signals. These simulations were evaluated on a hybrid computer which had an analog white noise generator and the differential equations of the HH model were solved by electrical analog components such as integrators and multiplicators. On digital computers differential equations are solved by time discretization. The noise signal changes its value also in time steps which may differ from the integration step. Thus the impact of noise on the spiking rate depends on noise amplitude and the time step for the noise transmission. Additionally, there is an essential difference how noise effects a single patch of membrane simulated by a local HH model and a conductance based cable model (multi-compartment model) of a nerve fiber that is stimulated externally by a microelectrode.

The current noise approach of Rattay [8] for a nerve fiber with HH dynamics is based on the observation that (i) transmembrane current fluctuations are closely related with the number of sodium ion channels [29] and (ii) the impact of noise increases with the square root of the number of noisy elements and the square root of time [30]. In more detail, the effective noise current measured in μA is assumed to be proportional to the square root of the number of sodium channels within a compartment

$$I_{noise,n} = GAUSS \cdot knoise \cdot \sqrt{A_n \cdot g_{Na}} \tag{1}$$

where GAUSS is a Gaussian noise current term (mean = 0, σ = 1) that changes its value every 2.5 μs, knoise is a factor common to all compartments, $A_n$ denotes membrane area in $cm^2$, and $g_{Na}$ is the maximum sodium conductance which is proportional to the number of sodium ion channels. The rather large noise transmission time Dt = 0.0025 ms is a compromise of computational costs and accuracy.

In the following we show (i) how knoise has to be changed for other noise transmission times, (ii) which values of knoise fit with recordings of Verveen using the models HH1 and HH10 for non-myelinated and myelinated axons, respectively, and (iii) why the CRRSS model needs essentially higher knoise values than HH10.

## Methods

The presented results are based on a straight axon that is stimulated by a small spherical electrode in distance z of the axis of an axon (Fig 2C). The axon is segmented into cylindrical compartments and every compartment is electrically reduced to its center point and described by the inner-axonal currents to the neighbors and the currents across the membrane (Fig 2A and 2B). The same principle is applied for the non-myelinated axon, just by replacing the internodes with active compartments and holding compartment length constant.

Kirchhoff's law (the sum of all currents is zero) for the n-th compartment of the electric network of Fig 2B results in

$$\frac{d(V_{i,n} - V_{e,n})}{dt} C_{m,n} + I_{ion,n} + \frac{V_{i,n} - V_{i,n-1}}{R} + \frac{V_{i,n} - V_{i,n+1}}{R} = 0 \tag{2}$$

where the first two terms are capacitance current and ion current across the membrane whereas the next two terms describe the intracellular current flow to the left and right compartment. Introducing the reduced membrane voltage $V = V_i - V_e - V_{rest}$ ($V_i$, $V_e$ and $V_{rest}$ are the intracellular, extracellular and resting potential, respectively) leads to the following system of differential equations [2, 8, 31].

$$\frac{dV_n}{dt} = \left[ -I_{ion,n} + \frac{V_{n-1} - V_n}{R} + \frac{V_{n+1} - V_n}{R} \right]/C_{m,n} + \left[ \frac{V_{e,n-1} - V_{e,n}}{R} + \frac{V_{e,n+1} - V_{e,n}}{R} \right]/C_{m,n} \tag{3}$$

For the first (last) compartment Eqs 2 and 3 have a reduced form because of lack of neighbors with index n-1 and n+1, respectively. The cylindrical membrane surface $A_n = d^*length^*\pi$ of every compartment with diameter d has to be calculated to find $C_{m,n} = A_n^* c_{m,n}$ ($c_{m,n}$ is the specific membrane capacitance; note that the capacitance of N layers of membranes is proportional to 1/N). Intracellular resistance $R = 4\rho_i \Delta x/d^2 \pi$ with intracellular resistivity $\rho_i = 0.13$ kOhm.cm and $\Delta x$ as distance between two compartment centers. In order to calculate the extracellular potentials $V_e$, needed in Eq 3, an infinite homogeneous extracellular medium is assumed. The extracellular potential was calculated as

$$V_e = \rho_e \cdot I_{electrode}/4 \pi r \tag{4}$$

where r is the center to center distance between a compartment of interest and a spherical electrode. $\rho_e = 0.3$ kOhm.cm was assumed to be the mean resistivity of the extracellular medium (Fig 2C). In the evaluations units were used following the work of Hodgkin and Huxley: cm (length), ms (time), mV (voltage), μA (current), μF (capacitance), kOhm (resistance) [16].

The ion membrane current is governed by the gating mechanisms of specific voltage sensitive ion channels. It consists of two components

$$I_{ion} = A_n \cdot i_{ion,n} + I_{noise,n} \tag{5}$$

where $i_{ion}$ is the ionic membrane current density and $I_{noise,n}$ represents ion channel current fluctuations in active compartments. The effective noise current measured in μA is assumed to be proportional to the square root of the number of sodium channels according to Eq (1)

In compartments with passive membranes (internodes) the term $I_{ion}$ of Eqs 1 and 2 is a current with constant conductance $I_{ion} = g_m A_n V_n/N$, where membrane conductance $g_m$ is 1 mS/cm$^2$ and the number of insulating layers of cell membranes N was 40 for thin (d < 2 μm) and 80 for thicker axons.

The ion currents of active membranes were simulated with original Hodgkin Huxley kinetics [12] for non-myelinated axon and with 10 fold Hodgkin Huxley membrane conductance (HH10) both at a temperature of 28.9°C which fits AP duration recorded in cats at 37°C [28].

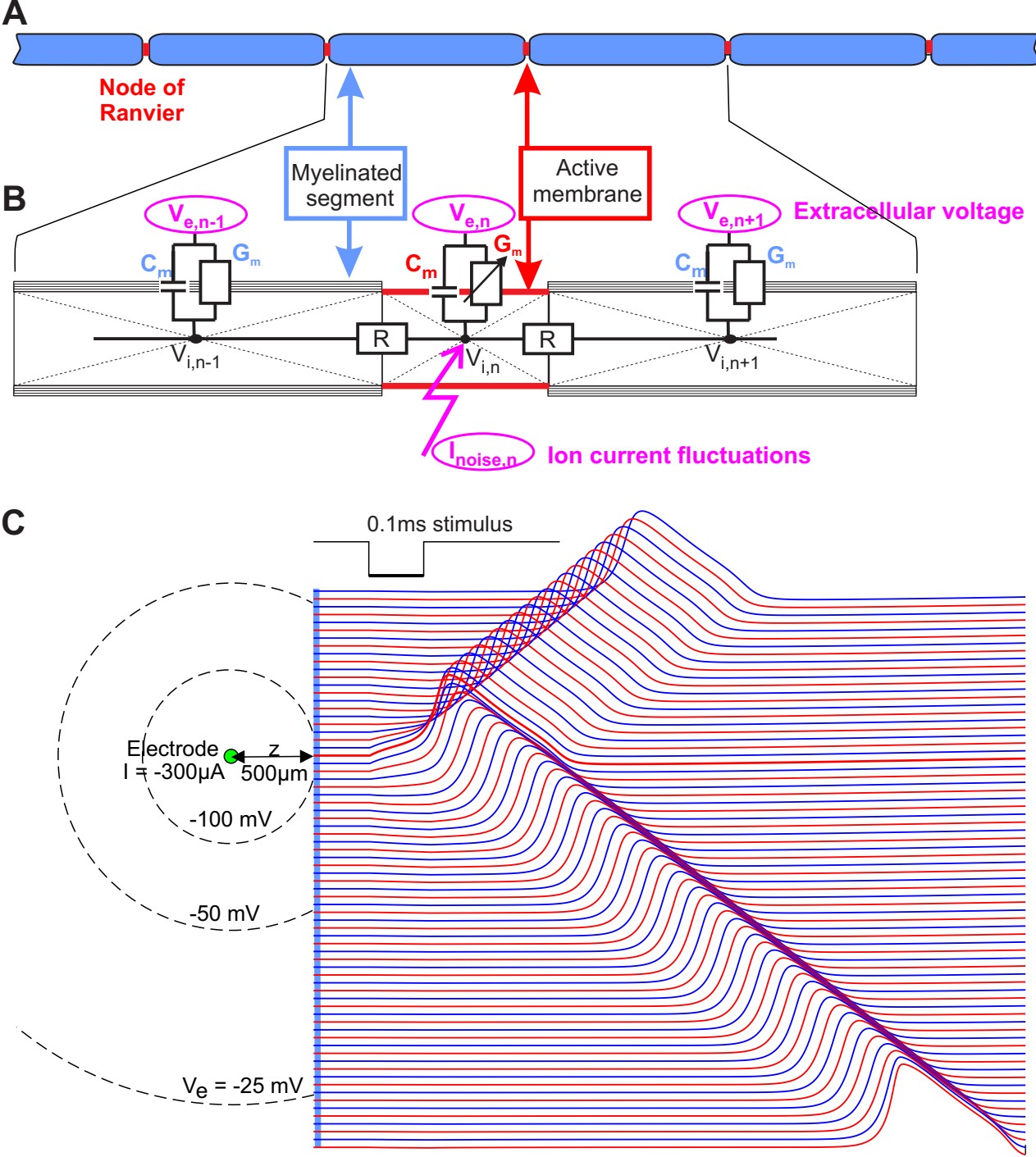

**Fig 2. Compartment model of a myelinated axon. A:** Geometry. The active compartments (nodes of Ranvier) are marked in red, the passive ones (internodes) in blue. **B:** Corresponding electrical network model. Independent noise currents $I_{noise,n}$ are added to the active compartments. The stimulating current from the electrode defines the extracellular potential ($V_e$) which causes a change of intracellular potential ($V_i$) depending on capacitance $C_m$ and conductance $G_m$ of the membrane and the intracellular resistance R. All influences from outside are marked via ellipses in magenta. **C:** Top view of the electrode and axon with equipotentials (left) and voltage profiles of active (red) and passive (blue) compartments (right). The excitation is initiated at the NoR closest to the electrode (thick red line). The uppermost 30 (of 101) compartments are not shown; simulation without noise.

**Table 1. Model parameters and formulas for the opening and closing rates of the ion channels.**

| | HH | HH10 | CRRSS |
|---|---|---|---|
| $\alpha_m$ | $\frac{2.5-0.1V}{\exp(2.5-0.1V)-1}$ | $\frac{2.5-0.1V}{\exp(2.5-0.1V)-1}$ | $\frac{97+0.363V}{\exp\left(\frac{31-V}{5.3}\right)+1}$ |
| $\beta_m$ | $4\cdot\exp\left(-\frac{V}{18}\right)$ | $4\cdot\exp\left(-\frac{V}{18}\right)$ | $\frac{\alpha_m}{\exp\left(\frac{V-23.8}{4.17}\right)}$ |
| $\alpha_n$ | $\frac{0.1-0.01V}{\exp(1-0.1V)-1}$ | $\frac{0.1-0.01V}{\exp(1-0.1V)-1}$ | |
| $\beta_n$ | $0.125\cdot\exp\left(-\frac{V}{80}\right)$ | $0.125\cdot\exp\left(-\frac{V}{80}\right)$ | |
| $\alpha_h$ | $0.07\cdot\exp\left(-\frac{V}{20}\right)$ | $0.07\cdot\exp\left(-\frac{V}{20}\right)$ | $\frac{\beta_h}{\exp\left(\frac{V-5.5}{5}\right)}$ |
| $\beta_h$ | $\frac{1}{\exp(3-0.1V)+1}$ | $\frac{1}{\exp(3-0.1V)+1}$ | $\frac{15.6}{1+\exp\left(\frac{24-V}{10}\right)}$ |
| $V_{rest}$ ($mV$) | -70 | -70 | -80 |
| $V_{Na}$ ($mV$) | 115 | 115 | 115 |
| $V_K$ ($mV$) | -12 | -12 | |
| $V_L$ ($mV$) | 10.6 | 10.6 | -0.01 |
| $g_{Na}$ ($mS/cm^2$) | 120 | 1200 | 1445 |
| $g_K$ ($mS/cm^2$) | 36 | 360 | |
| $g_L$ ($mS/cm^2$) | 0.3 | 3.0 | 128 |
| $c$ ($\mu F/cm^2$) | 1.0 | 1.0 | 2.5 |
| T (°C) | 28.9 | 28.9 | 37 |
| $\rho_e$ ($kOhm.cm$) | 0.3 | 0.3 | 0.3 |
| $\rho_i$ ($kOhm.cm$) | 0.13 | 0.13 | 0.13 |
| active compartment length ($cm$) | 5 d*) | 0.00025 | 0.00025 |
| passive compart-ment length ($cm$) | | 100 d | 100 d |
| number of compartments | 201 | 101 | 101 |

Note, V describes the reduced membrane voltage, which appears according to Hodgkin and Huxley as V = 0 for the resting state. The adjusted physical voltage is V +$V_{rest}$ in all models.

*) this length was increased with diameter; for d = 0.0001 cm it was 0.0005 cm.

For comparisons with the measurements of Verveen, we also used the CRRSS dynamics at 37°C to model the nodes of Ranvier, for details see Table 1 and [12, 16, 31]. We used 201 and 101 compartments for non-myelinated and myelinated axons, respectively and placed the electrode above the center of the axon. The system of ordinary differential equations was computed in C++ using the backward Euler method with time steps of 2.5 μs. Cathodic 0.1ms pulses were used for all simulations.

The HH model describes the ion current densities in μA/cm² as sum of sodium, potassium and leakage currents:

$$I_{ion} = g_{Na}m^3h(V - V_{Na})+g_Kn^4(V - V_K) + g_L(V - V_L) \tag{6}$$

where the gating probabilities m, n, h are defined by voltage dependent channel opening rates α and closing rates β (see Table 1 for details)

$$\frac{dm}{dt} = [-(\alpha_m + \beta_m)m + \alpha_m]k \tag{7}$$

$$\frac{dh}{dt} = [-(\alpha_h + \beta_h)h + \alpha_h]k \tag{8}$$

$$\frac{dn}{dt} = [-(\alpha_n + \beta_n)n + \alpha_n]k \tag{9}$$

the temperature factor $k = 3^{(0.1T-6.3)}$ with value k = 12, corresponding to T = 28.9˚C, was used. k = 1 is for the original experimental temperature of 6.3˚C.

The CRRSS model is similar to the HH model with the same Eqs (7) and (8) for ion channel gating, but without any potassium current; temperature factor $k = 3^{0.1T-3.7}$; k = 1 at T = 37˚C. The model describes the ion currents in the NoR based on experiments of a rabbit axon [31].

$$I_{ion} = g_{Na}m^2h(V - V_{Na}) + g_L(V - V_L) \qquad (10)$$

To determine RS, the intensity range from 0.6*threshold to 1.6*threshold was evaluated at 100 points, for which 1000 stochastic runs were performed for each individual intensity point. The corresponding spiking probability was determined as ratio (number of spikes)/1000. The RS was determined as the coefficient of variation of the approximating Gaussian distribution.

## Results

In neuroprosthetics firing efficiency, equivalent with firing probability, is an important control parameter that can be easily found for a stimulating pulse train as the ratio of elicited APs and the number of pulses. The sigmoidal increase of the firing probability with stimulus intensity is often quantified either by the dynamic range (DR) of recorded data or its relative spread (RS) (Fig 3). DR is the range of intensities where the spiking probability increases from 10% to 90% [21]. RS is based on experiments of Verveen who analyzed the stochastisity of axon thresholds and reported that such recorded sigmoid curves fit well with the cumulative distribution function of the normal distribution [22, 23]. A mathematical explanation for the relationship normalized DR = 2.56 RS is in the S1 Appendix.

The hypothesis of a linear relationship between the stochasticity measure RS and the common knoise factor was tested with the HH10 model by doubling knoise (Fig 4). The situation of Fig 4 is the same as shown in Fig 2C, but adding a noisy current to every active compartment enables the adjustment of knoise to experimental data.

### Adjustment of knoise to diameter dependent measurements of Verveen

Verveen reported a linear relationship between log RS and log d

$$\log RS = -1.5 - 0.8^*\log d \qquad (11)$$

(where diameter d is in μm) for a fit of his axon stimulation experiments that show how RS decreases with axon diameter [23]. In his approach Verveen made no distinction between myelinated and non-myelinated fibers and he did not investigate the electrode distance to the target fiber. We tested which knoise values for Dt = 0.0025ms are close to Verveen's approach for myelinated axons (d = 1 and 10 μm) and non-myelinated axons (d = 1, 10, 100 μm). According Verveen's Eq (11) RS values for d = 1, 10, 100 μm are 3.16, 0.50 and 0.08%. Quite similar log-log relations result for myelinated axons (Fig 5)

$$\text{HH10, } knoise = 0.00042: \quad \log RS = -1.51 - 0.76^*\log d \qquad (12)$$

$$\text{CRRSS, } knoise = 0.0038: \quad \log RS = -1.54 - 0.80^*\log d \qquad (13)$$

and non-myelinated axons

$$\text{HH1, } knoise = 0.0038: \quad \log RS = -1.48 - 0.83^*\log d \qquad (14)$$

The approximation quality to Verveen's formula is primarily determined by knoise and root of maximum sodium conductance times membrane area (Eq 1), while the distribution of X seems to play only a minor role. Instead of assuming X to be normally distributed, let X now be uniformly distributed on the interval [−1,1].

For the same values of knoise, regression equations result under the assumption of uniform distribution with deviations in the coefficients in the order of 1%:

$$\text{HH10, knoise} = 0.00042: \quad \log RS = -1.53 - 0.77^*\log d \tag{12U}$$

$$\text{CRRSS, knoise} = 0.0038: \quad \log RS = -1.51 - 0.79^*\log d \tag{13U}$$

$$\text{HH1, knoise} = 0.0038: \quad \log RS = -1.47 - 0.82^*\log d \tag{14U}$$

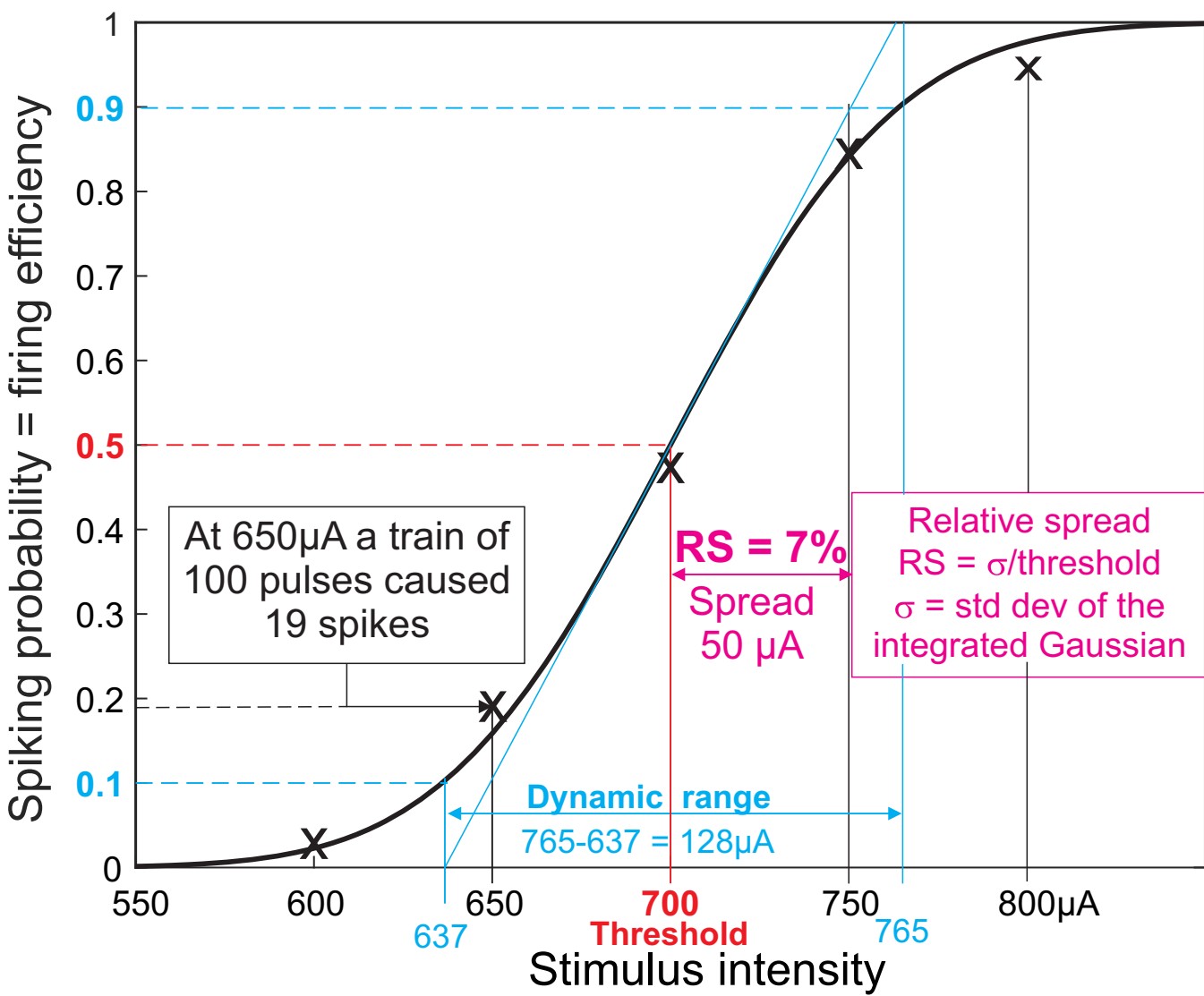

**Fig 3. Relationship between firing efficiency, relative spread (RS) and dynamic range (cyan).** The spread is plotted in magenta. The black curve is the Gaussian fit for five spiking probabilities (marked with x) with intensities 600, 650, 700, 750 and 800 μA. 700 μA represents the threshold value, i.e. the one in which a spike occurs in 50% of the stimulus trials. The dynamic range (normalized to threshold) corresponds to 2.56 times RS.

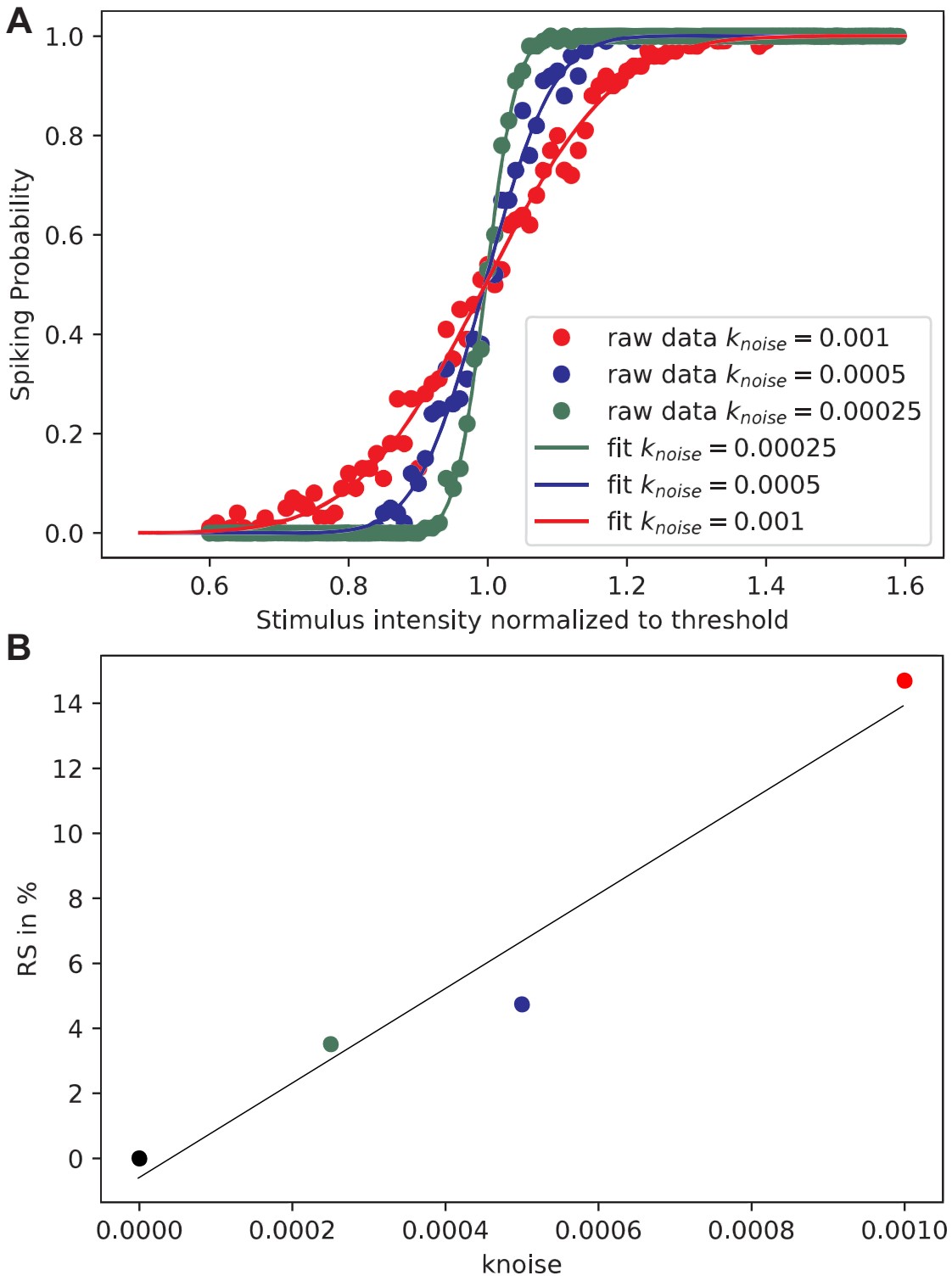

**Fig 4. RS increases linearly with knoise. A:** Spiking probability functions for three knoise values. Numerical evaluation was performed at 100 intensities, from 0.6 threshold to 1.6 ·threshold in 0.01 steps. For each point 500 runs were performed to estimate the spiking probability, then a cumulative Gaussian distribution was fitted. Electrode distance 500μm, myelinated axon d = 1 μm. **B:** RS values from A as functions of knoise underlines the assumption that for small knoise values RS is proportional to knoise. Linear fit: RS = 14458*knoise− 0.588, $R^2$ = 0.956.

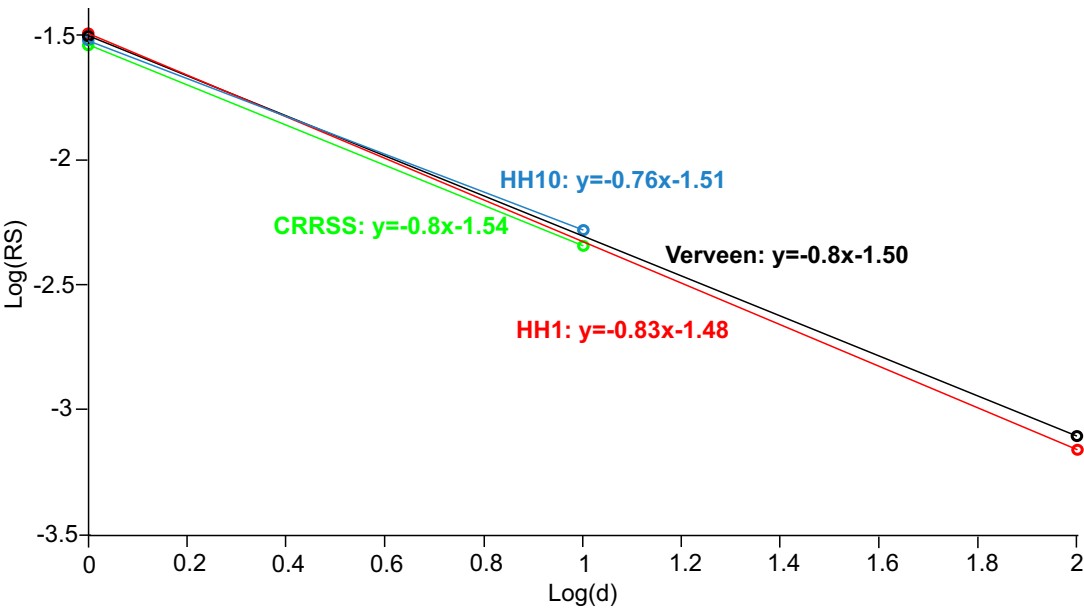

**Fig 5. Comparison of HH10, HH1, and CRRSS at adjusted knoise with Verveen's formula (black).** The CRRSS model of the myelinated fiber with knoise = 0.0038 (green) shows exactly the same slope of the straight line as Verveen, but slightly underestimates the vertical displacement. The HH10 model (knoise = 0.00042) of the myelinated fiber (blue) and the HH1 model (knoise = 0.0038) of the unmyelinated fiber (red) show slight deviations in slope and vertical displacement. Myelinated axon models were calculated with idealized internodes assuming $G_m = 0$ and $C_m = 0$ in Fig 2B.

## A formula for parameter adjustment for different noise transmission times

A transmission time Dt = 2.5 µs for the noise values may be not ideal for some investigations and therefore we present a formula for other time steps. Our hypothesis was that the impact of noise on RS increases with the square root of Dt. By systematically doubling the noise for axons (with ideal internodes $g_m = c_m = 0$), we confirmed an increase in RS in the range of 40–43% (Table 2), which includes the theoretically predicted increase value of $\sqrt{2}$ that is 41,42%.

**Table 2. Change of RS by doubling noise transmission time Dt.**

| Dt (ms) | d (µm) | RS (%) | Change of RS (%) |
|---|---|---|---|
| 0.000625 | 2 | 4.69 | |
| 0.00125 | 2 | 6.68 | 42.45 |
| 0.0025 | 2 | 9.38 | 40.41 |
| 0.005 | 2 | 13.27 | 41.47 |
| 0.01 | 2 | 18.82 | 41.85 |
| 0.000625 | 1 | 6.95 | |
| 0.00125 | 1 | 9.94 | 43.07 |
| 0.0025 | 1 | 13.98 | 40.58 |
| 0.005 | 1 | 19.63 | 40.34 |
| 0.01 | 1 | 32.99 | **68.03** |

For doubling Dt the theory predicts an increase of $\sqrt{2}$ that is 41,42%. This principle holds as long as Dt is not too large. For the last case (bold value), d = 1µm and Dt = 0.01ms, the rule did not hold since Dt is too coarse and thinner fibers are significantly noisier than thicker ones. Dt > 0.0025ms is generally not recommended. Simulated for electrode distance z = 1 mm, knoise = 0.001, HH10 model, 1000 runs per case.

Related to the standard value of Dt = 0.0025ms (Rattay 2000, Rattay et al. 2001) the following rule of thumb results for the conversion to a "new" time step Dt

$$k_{noise}(Dt) = \sqrt{\frac{0.0025}{Dt}} \cdot k_{noise}(0.0025) \tag{15}$$

## RS decreases with electrode distance

The closer the electrode is brought to the target fiber the less area is primarily stimulated. Consequently, less ion channels are involved to initiate the AP and a reduction of RS is expected when the electrode is moved towards the axon. In fact, it can be seen that the RS becomes smaller the shorter the electrode distance is chosen (Fig 6).

More details are found in Table 3. Our starting value of knoise = 0.00125 was from a previous study that fits reported RS values of feline auditory nerve fibers [24]. The upper part of Table 3 demonstrates that the CRRSS model needs about 10-fold knoise values of HH10 for similar RS and, second, the chosen knoise (0.00125) in the HH10 model leads to a significantly larger RS compared to Verveen's data. Since knoise is linearly related to RS adjusting the noise factor results in a good match to Verveen's formula for all cases tested (Table 3). For myelinated axons an about tenfold higher knoise value is recommended for the CRRSS model (knoise = 0.0038) versus the HH10 model (knoise = 0.00042), while knoise = 0.0038 as well is a good HH1 model fit for non-myelinated axons (Fig 5).

## Discussion

Our simple model approach is in good agreement with Verveen's measurements if the knoise parameter is individually adjusted for each of the membrane models (Fig 5, Eqs 12–14). Surprisingly, these knoise values for myelinated axon models differ by one order of magnitude for HH10 (knoise = 0.00042) and CRRSS (knoise = 0.0038) although they have similar maximum sodium conductances (gNa = 1200 mS/cm$^2$ in HH10 and gNa = 1445 mS/cm$^2$ in CRRSS). Interestingly, the threshold of the CRRSS axon is also much higher than that of HH10

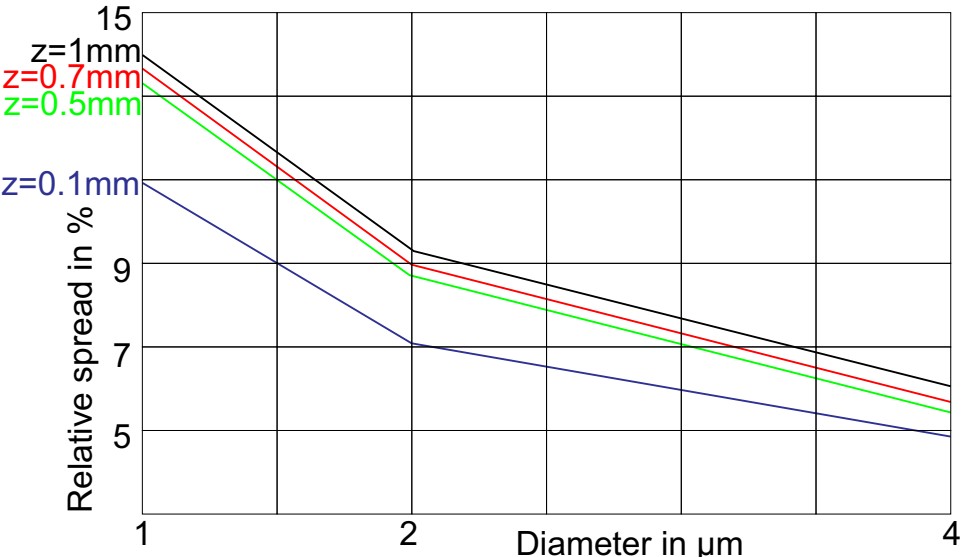

**Fig 6. RS as function of axon diameter for different electrode distances z from a myelinated fiber.** NoR length 2.1 µm, internode length 100*diameter, HH10.

**Table 3. Thresholds and RS for the HH1, HH10 and CRRSS model.**

| Diameter (μm) | Electrode distance (μm) | Threshold HH10 (μA) | Threshold CRRSS (μA) | knoise | RS HH10 (%) | RS CRRSS (%) | RS CRRSS (knoise x10) (%) |
|---|---|---|---|---|---|---|---|
| | | | Myelinated axon, HH10 vs. CRRSS | | | | |
| 1 | 200 | -28.4 | -145.27 | 0.00125 | 9.86 | 1 | 9.81 |
| 10 | 2000 | | -1452.7 | 0.00125 | 2.85 | 0 | 3.04 |
| 1 | 2000 | | | **0.00042** | **3.08** | | |
| 10 | 2000 | | | **0.00042** | **0.53** | | |
| 1 | 2000 | | | **0.0038** | | **2.91** | |
| 10 | 2000 | | | **0.0038** | | **0.46** | |

| Diameter (μm) | Electrode distance (μm) | Δx (μm) | Threshold (μA) | knoise | RS HH1 (%) | RS HH1 (knoise x3) (%) | RS HH1 (knoise x10) (%) |
|---|---|---|---|---|---|---|---|
| | | | Non-myelinated axon HH1 | | | | |
| 1 | 200 | 5 μm | -329.35 | 0.00125 | 1.07 | **3.18** | 11.12 |
| 10 | 2000 | 50 μm | -20041 | 0.00125 | 0.24 | **0.52** | 2.01 |
| 100 | 2000 | 500 μm | -3308 | 0.00125 | 0.02 | **0.07** | 0.36 |

Bold values are close to the approach of Verveen. Compartment length of the non-myelinated axon was increased for larger diameters. Threshold calculations without noise.

(Table 3). Obviously, the high threshold of the CRRSS model demands for related high noisy currents in order to reach the same noisiness as the data of Verveen. Comparison of the thresholds 329 vs. 28.4 μA for 1 μm diameter axons without and with myelin (HH1 vs. HH10, Table 3) is again of help to understand why the non-myelinated axon (HH1) fits Verveen's approach with the high knoise value of 0.0038.

Several other models are available for myelinated and non-myelinated axons [17, 32–34]. The models differ in capacitance $C_m$ and length of NoR by a factor up to 3 which is expected to require for a correction factor. In all cases it is easy to find a knoise value that fits with Verveen's data because of the linear relation between knoise and RS. This linear relationship results in a vertical shift of the line in the log-log plot of Fig 5 if a correcting factor is applied. Therefore, it is recommended to fit knoise for the diameter of interest, e.g. where spiking probability as a function of stimulus intensity is gathered from experiments. Spike probability data have been reported, e.g. from auditory nerve fibers and retinal ganglion cell axons [21, 35–37].

The RS of a human auditory nerve fiber during cochlea implant stimulation is of special interest as the spiking rate carries the loudness information of sound but the artificially generated spiking rate is controllable in the rather small intensity window called dynamic range which is, normalized to threshold, 2.56 times RS (Fig 3, S1 Appendix). The neural anatomy is a bit more complicated as the involved cells consist of quite different segments, a thin myelinated part followed by a thin non-myelinated segment, a non-myelinated soma and a thicker myelinated axon [38]. All these segments are close to the stimulating electrode because of a bending pathway of the neural structure [39] and thus RS depends on the spike initiation site which is sensitive to the electrode placing [40]. In persons with severe hearing deficits, the diameter of the peripheral part of an auditory nerve fiber can be reduced to 0.3 μm while the axon after the soma has a diameter of 4 μm [41]. For this example, the RS ratio is 6:1 for the parts before and after the soma. Similar situations of a non-homogeneous geometry consisting of dendrites, soma and axon, are reported for retinal ganglion cells where microelectrodes are placed close to the soma and spiking probability was plotted as function of stimulus intensity [13, 42].

Most of recent modeling work on extracellular nerve stimulation uses a two-step procedure: (i) determination of the electric potential induced by a stimulating electrode in a biological structure and (ii) prediction of its consequences for single-neuron responses [43, 44]. However, ignoring the stochastic current fluctuations across the cell membrane in this approach hinders the prediction of the spiking efficiency seen in experiments (Figs 1 and 3).

In order to show the impact of ion current fluctuations we have investigated the loudness discrimination with cochlear implants in a forthcoming study using a combined HH1 and HH10 model [24]. As mentioned above, the diameter of an auditory nerve fiber is much smaller in its peripheral part than after the soma. As a consequence, the simulated RS is about five times larger if the implanted electrode is close to the fiber ending in comparison to a position close to the center of the cochlea where it elicits spikes in the thick axon. As loudness perception, coded by the spiking rates of the stimulated fibers, depends on the dynamic range, we predict better loudness discrimination for electrodes close to the fiber endings [24]. This effect agrees with a study on intensity discrimination of 14 cochlear implant users with two types of electrode placements: subjects with the electrodes close to the cochlea center (far from fiber ending) had an average of only 9 discriminable intensity steps as compared to 23 steps in the other group [45]. Note that the classical nerve fiber model is useful for threshold calculations but is not of help to find the dependence of spiking rate on stimulus intensity.

The presented modelling approach is based on the observation that ion current fluctuations in a NoR are dominated by sodium ions [29]. Furthermore, it can be assumed that the amplitude of the noise current depends on the open probability of the channels and thus the amplitude of the noisy transmembrane currents is quite small in the resting state. However, for simplicity, but also because of lack of precise data, the root-mean-square (RMS) amplitude of the noise current is constant in every compartment of our model. This assumption overestimates the fluctuations in membrane voltage close to the resting state; compare noisy membrane voltage before and after the AP in the blue part of Fig 1.

Contrary to our simplification of constant RMS amplitudes, the noise term was simulated to increase in an exponential way with transmembrane voltage [9], which raises the technical effort. These technical adjustments also showed the negative linear trend between log(RS) to log(diameter) and the values were of similar magnitude as shown in Eq 11 and Fig 5. Another approach is to simulate the ion channels individually by Markov processes [46] and thus account for the random opening and closing behavior more accurately, whereby there are also gradations in sophistication, ranging from two-rate models to multi-rate models. Works based on stochastic modeling of single sodium channels also showed the decrease of RS with increase of excitable area [11, 25]. Their results are qualitatively transferable to our method because the membrane area is directly proportional to the number of sodium channels.

Computer simulation is a powerful tool to investigate the sensitivity of each model parameter. Here we have shown that RS increases with the distance of the electrode (Fig 6) and the electrode was a point source just above a NoR in an infinite large homogeneous medium with the ground far away. There are a lot of other parameters, models and scenarios, which can be analyzed with the presented method. As an example, high frequency stimulation is quite sensitive to noise and therefore ion current fluctuations should be included in modeling studies for neuroprosthetic applications that are based on such techniques [47–50]. While in these cited studies long regular axons are stimulated other modeling work is concerned with non-homogenous axons that include low threshold sodium channels in the axon initial segment of retinal neurons or pyramidal cells [6, 51]. In such studies on low current stimulation the presented method is recommended to obtain more insight about the population of excited cells around an active electrode, e.g., during microstimulation of the retina or cortex [52].

## Supporting information

**S1 Appendix.**
(DOCX)

## Author Contributions

**Conceptualization:** Frank Rattay.

**Formal analysis:** Frank Rattay, Thomas Tanzer.

**Investigation:** Frank Rattay, Thomas Tanzer.

**Supervision:** Frank Rattay.

**Validation:** Thomas Tanzer.

**Visualization:** Frank Rattay, Thomas Tanzer.

**Writing – original draft:** Frank Rattay, Thomas Tanzer.

**Writing – review & editing:** Frank Rattay, Thomas Tanzer.

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
