## [Decision Letter · Decision Letter 0]

24 Nov 2021

PONE-D-21-31403A simple model considering spiking probability during extracellular axon stimulationPLOS ONE

Dear Dr. Rattay,

Thank you for submitting your manuscript to PLOS ONE. After careful consideration, we feel that it has merit but does not fully meet PLOS ONE’s publication criteria as it currently stands. Therefore, we invite you to submit a revised version of the manuscript that addresses the points raised during the review process.

We look forward to receiving your revised manuscript.

Kind regards,

Jun Ma, Dr.

Academic Editor

PLOS ONE

Journal Requirements:

“This work was supported by the Austrian Science Fund (FWF), grant no. 29650.”

 Reviewer #1: A simple model considering spiking probability and noise was investigated in the present MS. Due to that too many information were not presented or the descriptions were too professional, I can not judge the importance of the present paper and how the results are acquired. For example, what’s the importance of the three points of the results, and how the results of tables or figures are acquired? And even no a whole model for the nerve fiber is provided. The nerve fiber consists of nodes of Ranvier and internodes were considered. What’s the model for nodes of Ranvier, what’s the model for internode, which were not stated clearly. In general, if the behavior of the nerve fiber is near the threshold, the noise will play important roles in modulating the generation of action potential. Both the membrane potential and noise type and intensity influence the generation of action potential. If the models of the nodes of Ranvier and internode are different, the results are complex due to different thresholds for the Ranvier and internode. In addition, the distance or number of Ranvier are also important. However, the results for the action potentials at nodes of Ranvier and internode, changes of action potential along with the distance, changes of the action potential with noise were not presented. Why the standard normal distribution? How the results of tables were acquired? Another sense for the present MS is too “specialized”, which is based on the results of some references. It is difficult to read for most of the readers, due to the contents and the expressions. In summary, the present paper maybe important, but which has not been presented in the abstract, introduction and results. If the MS can not be improved to a very large extent, I do not suggest the MS to be published.

Reviewer #2: The paper looks very raw, like a technical report. Though the underlying investigation is interesting. The main outcome of the paper is that the proposed model well matches the measurements if its parameters are fitted to the data. However, to understand this, the reader has to study the whole manuscript. It is clear that the paper should be reorganized. The main idea should be placed into the abstract and the introduction, the tables should be removed to supplementary and/or replaced by charts, the comparison of the proposed model and existing ones should be made explicitly. After a complete revision of the manuscript it should be studied more carefully for some minor issues.

Please, provide reproducibility of the results: all parameters should be put together. The methods and techniques for parameter fitting should be explicitly declared.

---

## [Author Response · Author response to Decision Letter 0]

10 Dec 2021

All requests on funding are answered in the cover letter

---

## [Decision Letter · Decision Letter 1]

3 Jan 2022

PONE-D-21-31403R1A simple model considering spiking probability during extracellular axon stimulationPLOS ONE

Dear Dr. Rattay,

Thank you for submitting your manuscript to PLOS ONE. After careful consideration, we feel that it has merit but does not fully meet PLOS ONE’s publication criteria as it currently stands. Therefore, we invite you to submit a revised version of the manuscript that addresses the points raised during the review process.

We look forward to receiving your revised manuscript.

Kind regards,

Jun Ma, Dr.

Academic Editor

PLOS ONE

Reviewer #1: The revised MS has been improved to a large extent. I suggest the MS to be published after revisions.

(1) The author claim that they consider an important factor which is often ignore in other investigations. The results acquired with their model close match the experimental results. The author should present the shortage of the results acquired by other models and compare their results and other results. In addition, the significance or potential application of this model or method should be presented.

(2) The figures should be improved, for example, the first letter should be capitalized. There are many errors in the References. Some recent references may be needed.

Reviewer #2: The problem of the paper comes from its idea. The authors consider that

1) the neuronal signal is binary, but there are many papers that subthreshold activity plays a significant role, see https://doi.org/10.1523/JNEUROSCI.19-24-10727.1999 for example (rather old paper!);

2) the external input is considered to be Gaussian (the 1st reviewers also addressed this problem, but the authors could not answer properly), but there are papers considering that this is not the case, see https://doi.org/10.1103/PhysRevE.97.060302 as an example.

These two issues are significant because the whole paper is based on them. If authors cannot address these questions, the paper should be rejected.

---

## [Author Response · Author response to Decision Letter 1]

10 Jan 2022

I have uploaded all files including the 'Response to Reviewers'

---

## [Decision Letter · Decision Letter 2]

16 Feb 2022

A simple model considering spiking probability during extracellular axon stimulation

PONE-D-21-31403R2

Dear Dr. Rattay,

We’re pleased to inform you that your manuscript has been judged scientifically suitable for publication and will be formally accepted for publication once it meets all outstanding technical requirements.

Kind regards,

Jun Ma, Dr.

Academic Editor

PLOS ONE

Reviewer #1: Considering that my commnets have been considered in the revised MS, I suggest the present to be published.

Reviewer #2: The paper is still too complex to read. However, since I read it three time, I can say now, that I mostly understand what is mentioned. I hope that other researchers will be more lucky and intelligent.

---

## [Editor Report · Acceptance letter]

21 Feb 2022

PONE-D-21-31403R2 

A simple model considering spiking probability during extracellular axon stimulation 

Dear Dr. Rattay:

I'm pleased to inform you that your manuscript has been deemed suitable for publication in PLOS ONE. Congratulations! Your manuscript is now with our production department. 

Kind regards, 

on behalf of

Dr. and Pro. Jun Ma 

Academic Editor

PLOS ONE